# TopoStreamer: Temporal Lane Segment Topology Reasoning in Autonomous Driving

## Abstract

Lane segment topology reasoning constructs a comprehensive road network by capturing the topological relationships between lane segments and their semantic types. This enables end-to-end autonomous driving systems to perform road-dependent maneuvers such as turning and lane changing. However, the limitations in consistent positional embedding and temporal multiple attribute learning in existing methods hinder accurate road network reconstruction. To address these issues, we propose TopoStreamer, an end-to-end temporal perception model for lane segment topology reasoning. Specifically, TopoStreamer introduces three key improvements: streaming attribute constraints, dynamic lane boundary positional encoding, and lane segment denoising. The streaming attribute constraints enforce temporal consistency in both centerline and boundary coordinates, along with their classifications. Meanwhile, dynamic lane boundary positional encoding enhances the learning of up-to-date positional information within queries, while lane segment denoising helps capture diverse lane segment patterns, ultimately improving model performance. Additionally, we assess the accuracy of existing models using a lane boundary classification metric, which serves as a crucial measure for lane-changing scenarios in autonomous driving. On the OpenLane-V2 dataset, TopoStreamer demonstrates considerable improvements over state-of-the-art methods, achieving substantial performance gains of **+3.0% mAP** in lane segment perception and **+1.7% OLS** in centerline perception tasks. Our code will be released.

## 1 Introduction

Perception serves as a crucial component in end-to-end autonomous driving (Li et al., 2024b; Yang et al., 2025b), providing essential road priors for planning. Existing HD map learning and lane topology reasoning methods primarily focus on frame-by-frame detection (Li et al., 2023b; Liao et al., 2022). This approach fails to account for instance consistency across consecutive frames, making it susceptible to missed detections due to occlusions and high-speed movements (Yuan et al., 2024). Such limitations significantly hinder continuous and smooth decision-making and maneuvers. To comprehensively leverage temporal information, streaming-based methods (Yuan et al., 2024; Wang et al., 2024b; Wu et al., 2025)

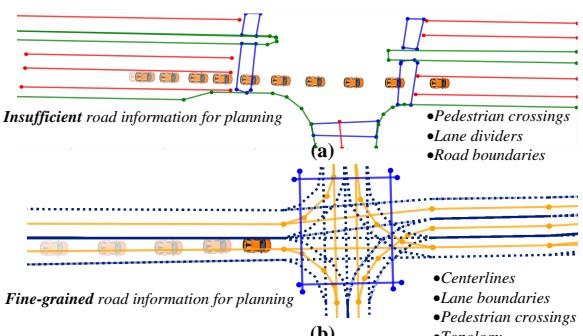

**Figure 1:** Comparsion between current streaming-based map learning methods (Yuan et al., 2024) and our TopoStreamer. TopoStreamer delivers more fine-grained road information through streaming perception of lane segments, which is vital for planning.

propose memory-based temporal propagation to establish long-term frame associations. Specifically, these approaches leverage the ego-vehicle pose to predict the probable positions of road instances in subsequent frames. However, these methods fail to capture sufficient road information for planning. This inspired us to introduce a temporal mechanism in lane topology reasoning, which

we can leverage perception and topology reasoning results from previous frames to predict current frame outcomes and capture fine-grained road information for planning (Jia et al., 2025). Fig. 1 demonstrates the comparison between our method and current streaming-based learning methods(Yuan et al., 2024). To the best of our knowledge, achieving this objective presents two primary challenges: **(1) Consistent positional embedding.** Current streaming-based methods exhibit deficiencies in their positional embedding design for stream queries. Furthermore, certain lane topology reasoning approaches (Li et al., 2023a;b) suffer from inconsistency between reference points and positional embedding updates. **(2) Temporal multiple attribute learning for lane segments.** Topology reasoning between lanes is highly sensitive to the precise localization of lane segments (Fu et al., 2025a) and projection errors make it challenging to maintain consistent localization and category of lane segments across temporal propagation.

To address the aforementioned challenges, we propose TopoStreamer, a novel temporal perception framework for lane segment topology reasoning. To strengthen positional embedding consistency, we augment the heads-to-regions mechanism (Li et al., 2023b) through dynamic explicit positional encoding across successive decoder layers. This design progressively injects updated positional information to enhance query updating with latest spatial learning. Furthermore, we introduce multiple streaming attribute constraints and a lane segment denoising module to reinforce temporal coherence and enable the learning of diverse patterns in lane segments. We also propose a new metric to evaluate the lane boundary classification accuracy, a measure for autonomous vehicle lane-changing decision-making systems.

**Contributions:** (1) We present TopoStreamer, a novel temporal lane segment perception method for lane topology reasoning in autonomous driving. (2) Three novel modules have been proposed, including streaming attribute constraints for lane segments in temporal propagation, a dynamic lane boundary positional encoding module to enhance positional learning, and a lane segment denoising module for the learning of diverse patterns in lane segments. (3) Extensive experiments conducted on lane segment benchmark OpenLane-V2 (Wang et al., 2024a) demonstrate SOTA performance of TopoStreamer in lane topology reasoning.

## 2 RELATED WORK

### 2.1 HD MAP AND LANE TOPOLOGY REASONING

Traditional high-definition (HD) map reconstruction primarily relies on SLAM-based methods (Zhang et al., 2014; Shan & Englot, 2018), which incur substantial costs in manual annotation and map updates. With recent advancements in bird's-eye view (BEV) perception and detection frameworks, offering improved efficiency and performance, the research focus has shifted towards vectorized HD map learning approaches. HDMapNet (Li et al., 2022b) generates HD semantic maps from multi-modal sensor data. However, extra post-processing is required to obtain vectorized representations. To generate vectorized map directly, VectorMapNet (Liu et al., 2023) predicts map elements as a set of polylines. The MapTR series (Liao et al., 2022; 2023) propose precise map element modeling and stabilizes learning via a hierarchical query-based anchor initialization mechanism. Unlike online HD map methods that primarily focus on drivable boundaries, our method concentrate on lane topology reasoning to perceive drivable trajectories (centerlines) and their topological relationships. STSU (Can et al., 2021) predicts an ordered lane graph to represent the traffic flow in the BEV. Subsequent research (Wu et al., 2023; Li et al., 2023a) has explored centerline topology using diverse model architectures on the OpenLane-V2 benchmark. To address endpoint misalignment issues in topology prediction, TopoLogic introduces dual constraints: distance-aware and similarity-aware optimization objectives. LaneSegNet (Li et al., 2023b) proposes lane segment perception to enhance the complete description of map. TopoPoint (Fu et al., 2025b) proposes Point-Lane interaction to learn accurate endpoints for reasoning. However, aforementioned methods overlook the potential benefits of temporal consistency for lane perception. In this work, we propose temporal-aware lane segment learning.

### 2.2 TEMPORAL 3D OBJECT DETECTION

In open-world scenarios, single-frame 3D detection faces challenges stemming from inaccurate pose estimation, occlusion, and adverse weather conditions. To overcome these limitations, recent ad-

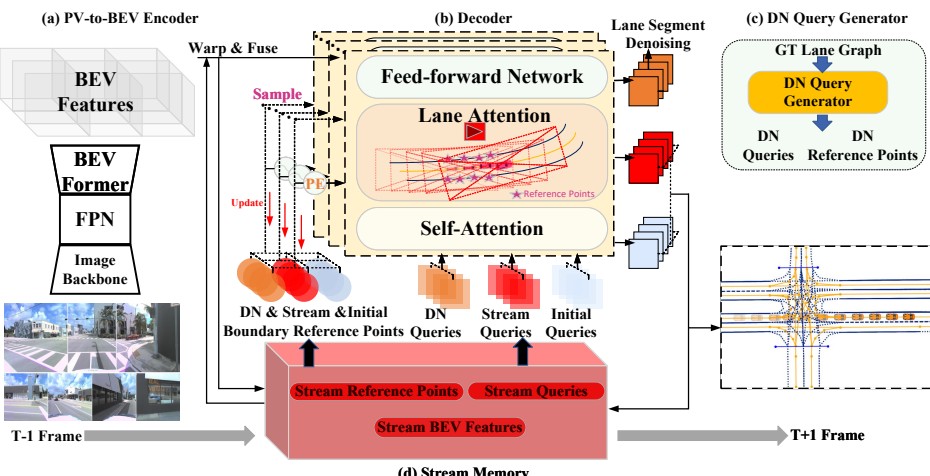

**Figure 2:** The overall architecture of TopoStreamer. It consists of four main components: **part (a)** PV-to-BEV encoder for extracting BEV features from multi-view images, **part (b)** transformer-based decoder enhanced with a dynamic lane boundary positional encoding module to improve up-to-date positional information learning, **part (c)** DN query generator for lane segment denoising, and **part (d)** stream memory that enables temporal propagation.

vancements have incorporated long-term memory to store different feature. BEVFormerv2 (Yang et al., 2023) and BEVDet4D (Huang & Huang, 2022) stack BEV features from historical frames. Sparse4D (Lin et al., 2022) and PETRv2 (Liu et al., 2022) design sparse fusion on images feature to avoid dense perspective transformation. StreamPETR (Wang et al., 2023) and Sparse4Dv3 (Lin et al., 2023) propagate historical information in query feature frame by frame. The temporal detection also shows impressive results in HD map learning. StreamMapNet (Yuan et al., 2024) proposes a streaming-based framework to warp and fuse the BEV features, and top-k reliable queries are selected to propagate. MapTracker (Chen et al., 2024) propose a tracking-based temporal fusion framework. It fuses the BEV and query features with distance strides to ensure extended-range consistency. SQD-MapNet proposes a denoising method for map elements to address the issue of information loss at the boundaries of BEV grid. Different from these methods, our distinctive contribution lies in the introduction of customized enhancements specifically designed for the more complex task of lane segment perception. This not only facilitates comprehensive road network understanding but also addresses the critical gap in positional embedding (PE) modeling within existing temporal map learning methods.

### 2.3 QUERY DENOISING

Adding noise and performing denoising during training has been proven to accelerate the training process and enhance the model's capabilities in both classification and regression. DN-DETR (Li et al., 2022a) introduces a denoising part apart from matching part as auxiliary supervision. DINO (Zhang et al., 2022) introduces a contrastive training to distinguish hard DN samples. SQD-MapNet (Wang et al., 2024b) proposes a stream query denoising to address the issue of map element truncation at boundaries caused by pose changes. To learn a comprehensive road network, we predict multiple attributes into a single lane segment query. To predict these attributes accurately, we design a tailored denoising learning strategy specifically for lane segments.

### 3 METHOD

Given surrounding multi-view images $\mathbf{I}$, our goal is to predict 3D position, class attributes and topology of lane segments. Each lane segment is composed of a centerline $\mathbf{L}^c = (\mathbf{P}, C)$, a left lane boundary $\mathbf{L}^l = (\mathbf{P}, T)$, and a right lane boundary $\mathbf{L}^r = (\mathbf{P}, T)$. $\mathbf{P}$ denotes an ordered set of points $\mathbf{P} = \{(x_i, y_i, z_i)\}|_{i=1}^M$, where $M$ is a preset number of points. In fact, we can obtain the boundary coordinates simply by predicting an offset and applying it to the centerline coordinates. $C$ indicates the lane segment class, which includes categories such as road lines and pedestrian crossings. $T$

denotes the boundary class, which can be dashed, solid, or non-visible. The connectivity topology is indicated by an adjacency matrix $\mathbf{A}$ (Can et al., 2021).

Table 1: Meaning of the notations in TopoStreamer

| Notation | Meaning |
|---|---|
| $\mathbf{L}^c, \mathbf{L}^l, \mathbf{L}^r$ | Center, left boundary and right boundary lines |
| $\mathbf{F}_{bev}, \mathbf{F}_{pe}, \mathbf{F}_{content}$ | BEV feature, positional embedding, and content embedding |
| $\mathbf{Q}$ | Queries |
| $\mathbf{D}, \mathbf{S}, \mathbf{I}$ | Denoising (DN), stream and initialized |
| $t$ | Time stamp T |
| $\mathbf{R_B}, \mathbf{R_C}$ | Boundary and center reference points |
| $C, T$ | Lane segment class and boundary class |
| $\mathbf{M}, \mathbf{A}$ | BEV semantic mask and adjacency matrix |
| $\mathbf{P} = \{(x_i, y_i, z_i)\}$ | An ordered set of points that forms a lane |
| $\Psi$ | Transformation matrix |

## 3.1 OVERALL ARCHITECTURE

The overall architecture of TopoStreamer is illustrated in Fig. 2. For clarity, some main notations in NLPF are displayed in Tab. 1. First, the surrounding multi-view images are processed by the PV-to-BEV encoder (Li et al., 2022c; He et al., 2016; Lin et al., 2017a) to generate BEV features $\mathbf{F}_{bev} \in \mathbb{R}^{H \times W \times C}$, where C, H, W represent the number of feature channels, height, and width, respectively. These current BEV features are then fused with past BEV features. A DN query generator provides denoising (DN) queries $\mathbf{Q^D} \in \mathbb{R}^{N \times C}$, DN center reference points $\mathbf{R_C^D} \in \mathbb{R}^{N \times 10 \times 3}$, and DN boundary reference points $\mathbf{R_B^D} \in \mathbb{R}^{N \times 10 \times 3}$. Next, a transformer-based decoder (Zhu et al., 2020) refines the DN, stream, and initialized queries $\{\mathbf{Q^D}, \mathbf{Q^S}, \mathbf{Q^I}\}$. The BEV features, along with DN, stream, and initialized boundary reference points $\{\mathbf{R_B^D}, \mathbf{R_B^S}, \mathbf{R_B^I}\}$, are subsequently fed into the lane attention (Li et al., 2023b) along with the corresponding queries for further processing. A dynamic lane boundary positional encoding module encodes these boundary reference points into positional embeddings $\mathbf{F}_{pe} \in \mathbb{R}^{N \times 10 \times C}$, injecting positional information into the queries layer by layer. Meanwhile, the DN, stream, and initialized center reference points $\{\mathbf{R_C^D}, \mathbf{R_C^S}, \mathbf{R_C^I}\}$ are utilized for prediction refinement (Zhu et al., 2020). The updated DN queries are used for lane segment denoising, while the stream and initialized queries are employed by the prediction heads to generate the lane graph. Additionally, past BEV features, queries and reference points are stored in a stream memory, enabling temporal propagation.

## 3.2 TEMPORAL PROPAGATION FOR LANE SEGMENT

Since lane segments remain stationary in geodetic coordinate while ego-vehicle poses change relative to them, we can utilize the detection results from the previous frame combined with the vehicle's ego-motion to establish reference positions for subsequent frame predictions (Yuan et al., 2024). First, we warp the BEV features from the past frame to fuse with the BEV features from the current frame by a Gated Recurrent Unit (GRU) (Chung et al., 2014):

$$\tilde{\mathbf{F}}_{bev}^t = \text{GRU}(\text{Warp}(\mathbf{F}_{bev}^{t-1}, \Psi), \mathbf{F}_{bev}^t) \tag{1}$$

where $\Psi$ denotes transformation matrix between two frames. Then the BEV features are stored in the stream memory for fusion in the next frame.

For query propagation across consecutive frames, we implement a learnable transformation through a MLP, which adaptively maps the top-k highest confidence queries from the previous frame to the current frame's coordinate system. Then, we can obtain the stream queries:

$$\mathbf{Q_t^S} = \text{MLP}(\text{Concat}(\mathbf{Q_{t-1}}, \Psi)) + \mathbf{Q}_{t-1} \tag{2}$$

where $\text{Concat}(\cdot)$ denotes the concatenate function and $\mathbf{Q}_{t-1}$ can be the stream and initialized queries from t-1 frame. DN queries and DN reference points are excluded from temporal propagation.

**Streaming Attribute Constraints.** Conventional approaches (Yuan et al., 2024; Wang et al., 2024b; Chen et al., 2024) typically apply transformation loss to the converted coordinates to facilitate the learning of coordinate transformation. Since lane segments inherently possess multiple attributes, maintaining their temporal consistency requires more sophisticated constraints than simple coordinate transformation loss. To address this, we develop a comprehensive set of streaming attribute constraints. We employ MLPs to predict lane segment coordinate $\tilde{\mathbf{L}}_t = \text{Concat}(\tilde{\mathbf{L}}_t^c, \tilde{\mathbf{L}}_t^l, \tilde{\mathbf{L}}_t^r)$ lane segment class $\tilde{C}_t$, boundary class $\tilde{T}_t$ and BEV semantic mask $\tilde{\mathbf{M}}_t$ from stream queries $\mathbf{Q}_t^\mathbf{S}$. Then, the streaming attribute constraints are represented as:

$$\begin{aligned}
\mathcal{L}_{coord}^{Stream} &= \mathcal{L}_{L1}(\tilde{\mathbf{L}}_t, \mathbf{L}_t) \\
\mathcal{L}_{cls}^{Stream} &= \mathcal{L}_{Focal}(\tilde{C}_t, C_t) + \mathcal{L}_{CE}(\tilde{T}_t, T_t) \\
\mathcal{L}_{mask}^{Stream} &= \mathcal{L}_{CE}(\tilde{\mathbf{M}}_t, \mathbf{M}_t) + \mathcal{L}_{Dice}(\tilde{\mathbf{M}}_t, \mathbf{M}_t) \\
\mathcal{L}_{Stream} &= \mathcal{L}_{coord}^{Stream} + \mathcal{L}_{cls}^{Stream} + \mathcal{L}_{mask}^{Stream}
\end{aligned} \tag{3}$$

where, for brevity, we omit the weights for each loss term. $\mathbf{L}_t$, $T_t$, $C_t$ and $\mathbf{M}_t$ are GT annotations transformed from T-1 frame to T frame. More details can be found in appendix.

**Lossless Streaming Supervision.** Existing approaches (Yuan et al., 2024; Wang et al., 2024b) utilize GT annotations from the past frame, transformed via pose estimation, to supervise the transformation loss for the subsequent frame. However, this method inevitably leads to information loss at BEV boundary regions as shown in (Wang et al., 2024b). To address this limitation, we track the unique IDs of positive instances matched through Hungarian assignment for stream queries, thereby providing lossless supervision. This is made possible by OpenLane-V2's provision of unique instance IDs. For datasets that do not provide IDs, we can also use mask matching (Chen et al., 2024) to identify ID associations across frames.

### 3.3 DYNAMIC LANE BOUNDARY PE

As shown in Fig. 3, current temporal approaches (Yuan et al., 2024; Wang et al., 2024b; Chen et al., 2024) neglect the learning of positional embeddings, leading to inaccurate spatial localization. Furthermore, a critical updating inconsistency exists in recent single-frame detection methods (Li et al., 2023b;a). Because the initialized positional embeddings remain static, while the reference points are updated layer by layer. In the temporal propagation process, when some initial queries are substituted with stream queries, combining them with static PE could result in incompatible feature integration due to mismatches. The static PE refers to the PE that is not updated between layers during the forward pass. Furthermore, existing

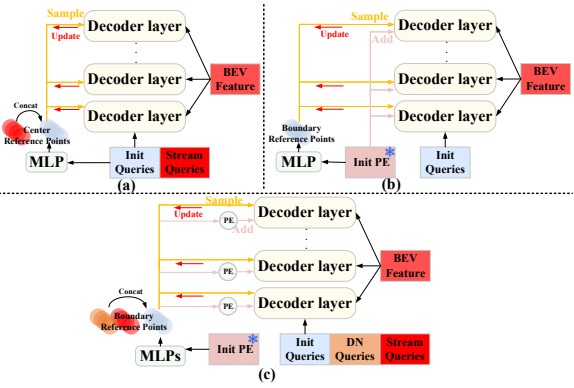

**Figure 3:** Comparison of PE: (a) current streaming-based approaches (Yuan et al., 2024), (b) recent single-frame detection methods (Li et al., 2023b), and (c) our proposed method.

methods (Liu et al., 2024) primarily focus on injecting positional encoding (PE) for the centerline. However, in lane segment recognition, this centerline-based PE injection can diminish the focus on positions within the boundary lines, which contradicts our goal of predicting multiple attributes for the entire lane segment area. To address these problems, we enhance the heads-to-regions sampling module (Li et al., 2023b) by a dynamic lane boundary PE modeling. We apply point-wise positional encoding (Liu et al., 2024) to the boundary reference points to generate positional embeddings. We duplicate the queries to align with the number of boundary reference points. After the self-attention, we combine the positional embeddings with the corresponding queries. Subsequently, the queries interact with and BEV features through lane attention (LA) (Li et al., 2023b) with boundary reference points sampling. Finally, a MLP is used to merge the duplicated queries. This process can be

represented as:

$$\mathbf{F}_{pe} = \{\text{PE}(\mathbf{P_i^B})\}$$
$$\tilde{\mathbf{Q}} = \text{MLP}(\text{LA}(\text{Duplicate}(\text{SA}(\mathbf{Q})) + \mathbf{F}_{pe}, \tilde{\mathbf{F}}_{bev}, \mathbf{R_B})) \tag{4}$$

where $\tilde{\mathbf{Q}}$ denotes the updated queries by this layer. Then, the reference points are refined through offset adjustments to enable more precise sampling. This refinement facilitates the injection of more accurate positional embeddings into the queries in subsequent decoder layers, thereby enhancing the learning of precise lane segment localization. For more implementation details regarding the decoder process, please refer to the appendix.

## 3.4 LANE SEGMENT DENOISING

The accuracy of topology prediction is highly dependent on the quality of lane detection. For instance, when noise causes misalignment between the endpoints of two lane segments that should be connected, it can significantly compromise the reliability of topological relationship inference. To address this, we introduce a denoising mechanism (Li et al., 2022a; Wang et al., 2024b) during training, which enables the model to learn from various noisy patterns. These noise patterns often lead to fragmented lane segments and positional shifts, thereby reducing the likelihood of correct topological associations. By learning to denoise, the model can recover the original positions and connectivity of lane segments, ultimately improving both detection robustness and topology inference performance. A detailed denoising example is provided in the supplementary material. In contrast to object or HD map detection (Li et al., 2022a; Wang et al., 2024b), which separately predicts bounding boxes or polylines along with their categories, lane segment perception involves a more complex set of attributes. These encompass centerlines, boundary coordinates, segment types, and downstream topological relationships. Consequently, our design of the denoising queries and objective loss function comprehensively accounts for this multifaceted nature.

Fig. 4 illustrates the generation of DN queries for lane segment perception. Initially, noise is introduced to the ground truth (GT) data. Then, the DN queries are obtained through content and positional embedding:

$$\mathbf{F}_{pe}^D = \text{MLP}(\{\text{PE}(\mathbf{R_C})\})$$
$$\mathbf{F}_{content}^D = \text{MLP}(\text{Concat}(\text{Emb}(C), \text{Emb}(T)))$$
$$\mathbf{Q^D} = \text{MLP}(\text{Concat}(\mathbf{F}_{pe}^D, \mathbf{F}_{content}^D)) \tag{5}$$

where Emb denotes embedding mapping operation.

Subsequently, we introduce a set of denoising losses to rectify noisy coordinates, category misclassifications, and topological errors arising from coordinate deviations in the DN queries. The denoising losses include L1 loss for position regression, cross-entropy and focal loss (Lin et al., 2017b) for classification and adjacency matrix prediction:

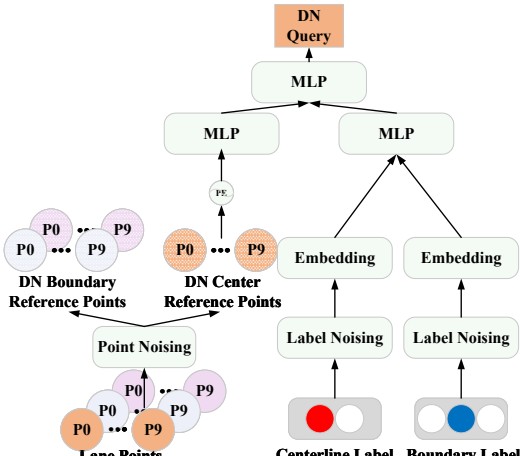

**Figure 4:** DN query for lane segment denoising.

$$\mathcal{L}_{denoise} = \mathcal{L}_{coord}^{DN} + \mathcal{L}_{cls}^{DN} + \mathcal{L}_{topo}^{DN} \tag{6}$$

where, for brevity, we omit the weights for each loss term. Details of DN losses can be found in appendix. This module enhances the learning of diverse patterns in lane segments, specifically correcting erroneous predictions caused by temporal projection errors.

## 3.5 TRAINING LOSS

The overall loss function in TopoStreamer is defined as follows:

$$\mathcal{L} = \alpha_1 \mathcal{L}_{ls} + \alpha_2 \mathcal{L}_{stream} + \alpha_3 \mathcal{L}_{denoise} \tag{7}$$

**Table 2:** Comparison with the state-of-the-arts on OpenLane-V2 benchmark on lane segment. All models adopt ResNet-50 as the backbone network and are trained for 24 epochs. [†]: Our enhanced model employ GeoDist strategy from TopoLogic (Fu et al., 2025a).

| Method | Venue | Temporal | mAP ↑ | $AP_{ls}$ ↑ | $AP_{ped}$ ↑ | $TOP_{lsls}$ ↑ | $Acc_b$ ↑ | FPS |
|---|---|---|---|---|---|---|---|---|
| MapTR (Liao et al., 2022) | ICLR23 | No | 27.0 | 25.9 | 28.1 | - | - | 14.5 |
| MapTRv2 (Liao et al., 2023) | IJCV24 | No | 28.5 | 26.6 | 30.4 | - | - | 13.6 |
| TopoNet (Li et al., 2023a) | Arxiv23 | No | 23.0 | 23.9 | 22.0 | - | - | 10.5 |
| LaneSegNet (Li et al., 2023b) | ICLR24 | No | 32.6 | 32.3 | 32.9 | 25.4 | 45.9 | 14.7 |
| TopoLogic (Fu et al., 2025a) | NIPS24 | No | 33.2 | 33.0 | 33.4 | **30.8** | - | - |
| Topo2Seq (Yang et al., 2025a) | AAAI25 | No | 33.6 | 33.7 | 33.5 | 26.9 | 48.1 | 14.7 |
| StreamMapNet (Yuan et al., 2024) | WACV24 | Yes | 20.3 | 22.1 | 18.6 | 13.2 | 33.2 | 14.1 |
| SQD-MapNet (Wang et al., 2024b) | ECCV24 | Yes | 26.0 | 27.1 | 24.9 | 16.6 | 39.4 | 14.1 |
| **TopoStreamer (ours)** | - | Yes | **36.6** | 35.0 | **38.1** | 28.5 | 50.0 | 13.6 |
| **TopoStreamer[†] (ours)** | - | Yes | 36.5 | **35.1** | 37.8 | 30.1 | **50.2** | 13.2 |

where the lane segment loss $\mathcal{L}_{ls}$ supervises predicted lane segments through Hungarian matching (Li et al., 2023b), while $\mathcal{L}_{stream}$ and $\mathcal{L}_{denoise}$ are loss specifically optimized for lane segments streaming and denoising. $\alpha_1$, $\alpha_2$ and $\alpha_3$ are hyperparameters.

# 4 EXPERIMENTS

We evaluate our method on multi-view lane topology benchmark OpenLane-V2 (Wang et al., 2024a). Since lane segment labels are exclusively available in **subsetA**, our validation is primarily conducted on this subset. The results on subsetB can be found in appendix.

## 4.1 DATASETS AND METRICS

**OpenLane-V2** (Wang et al., 2024a) is a widely-used dataset for lane topology reasoning. Its subsetA, re-annotated from Argoverse 2 (Wilson et al., 2023), provides enhanced details on traffic signals, centerlines, lane boundaries, and their topological relationships. This subset includes over 20,000 training frames and more than 4,800 validation frames, with each frame comprising 7 camera images at a resolution of 2048 × 1550.

**Metrics**. We evaluate our model on two tasks: lane segment and centerline perception. The lane quality are evaluated under Chamfer distance and Frechet distance under a preset thresholds of {1.0, 2.0, 3.0} meters.

**Table 3:** Comparison with the state-of-the-arts on OpenLane-V2 benchmark on centerline perception. All models adopt ResNet-50 as the backbone network and are trained for 24 epochs. Unlike other methods, TopoFormer[*] adopts a staged training strategy that utilizes a pretrained lane detector for topology reasoning training. While this leads to better detection performance, it offers only slight advantage in topology prediction.

| Method | Venue | Temporal | OLS ↑ | $DET_l$ ↑ | $TOP_{ll}$ ↑ |
|---|---|---|---|---|---|
| VectorMapNet (Liu et al., 2023) | ICML23 | No | 13.8 | 11.1 | 2.7 |
| STSU (Can et al., 2021) | ICCV21 | No | 14.9 | 12.7 | 2.9 |
| MapTR (Liao et al., 2022) | ICLR23 | No | 21.0 | 17.7 | 5.9 |
| TopoNet (Li et al., 2023a) | Arxiv23 | No | 30.8 | 28.6 | 10.9 |
| Topo2D (Li et al., 2024a) | Arxiv24 | No | 38.2 | 29.1 | 26.2 |
| TopoMLP (Wu et al., 2023) | ICLR24 | No | 37.4 | 28.3 | 21.7 |
| LaneSegNet (Li et al., 2023b) | ICLR24 | No | 40.7 | 31.1 | 25.3 |
| TopoLogic (Fu et al., 2025a) | NIPS24 | No | 39.4 | 29.9 | 23.9 |
| Topo2Seq (Yang et al., 2025a) | AAAI25 | No | 42.7 | 33.5 | 27.0 |
| TopoFormer[*] (Lv et al., 2025) | CVPR25 | No | 42.1 | 34.7 | 24.7 |
| StreamMapNet (Yuan et al., 2024) | WACV24 | Yes | 28.8 | 21.7 | 12.9 |
| SQD-MapNet (Wang et al., 2024b) | ECCV24 | Yes | 33.9 | 27.2 | 16.4 |
| **TopoStreamer (ours)** | - | Yes | **44.4** | **35.2** | **28.8** |

For lane segment, mAP computed as average of $AP_{ls}$ and $AP_{ped}$. $AP_{ls}$ and $AP_{ped}$ are used to estimate the quality of lane segment of road lines and pedestrian crossing, respectively. $TOP_{lsls}$ measures the performance of topology reasoning. We design a new metric $Acc_b$ to evaluate lane boundary classification accuracy, which can be referred in appendix. The metrics in cenerline perception are similar with those in lane segment. OLS (Wang et al., 2024a) is calculated between $DET_l$ and $TOP_{ll}$.

## 4.2 EXPERIMENTAL SETTINGS

We adopt a pre-trained ResNet-50 (He et al., 2016), FPN (Lin et al., 2017a) and BevFormer (Li et al., 2022c) to encode the images to BEV features. The BEV grid is 200 × 100, which the perception

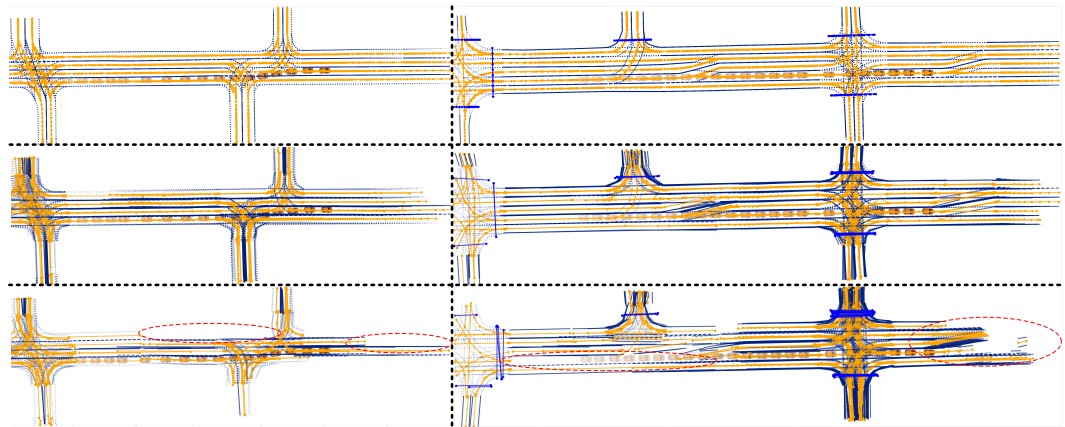

**Figure 5:** The temporally accumulated perception results are compared across GT, our TopoStreamer, and SQD-MapNet (Wang et al., 2024b). The images from top to bottom correspond to GT, TopoStreamer, and SQD-MapNet. Missing detections are highlighted with red color. For better viewing, zoom in on the image.

range is $\pm$ 50m $\times$ $\pm$ 25m. Our decoder is based on Deformable DETR, with the cross-attention replaced by lane attention (Li et al., 2023b). The number of layer is 6. We use 200 queries, with 30% allocated for temporal propagation. The centerline, left boundary line, and right boundary line are each represented as individual sets of 10 ordered points in our predictions. We select 8 boundary reference points (4 from the left boundary and 4 from the right) to generate PE. The number of DN groups is dynamically adjusted based on batch instances, while DN queries are fixed at 240. Positional noise is introduced via box shifting (Wang et al., 2024b) with a factor of 0.2, and labels have a 50% flip probability. Training is conducted for 24 epochs with a batch size of 8 on NVIDIA V100 GPUs, with the first 12 epochs using single-frame input to stabilize streaming training. The initial learning rate is $2\times10^{-4}$ with a cosine annealing schedule during training. AdamW (Kingma & Ba, 2015) is adopted as optimizer. The values of $\alpha_1$, $\alpha_2$, $\alpha_3$ are set to 1.0, 0.3 and 1.0, respectively. The confidence threshold for the adjacency matrix is set at 0.5, and all visualized segments must exceed a threshold of 0.3.

We re-train StreamMapNet (Wang et al., 2024b) and SQD-MapNet (Wang et al., 2024b), both of which predict 10 points for the left and right boundaries. The centerline is obtained by calculating the average positions of two boundaries.

### 4.3 MAIN RESULTS

**Results on Lane Segment** The results for lane segment are shown in Tab. 2. Compared with single-frame detection methods, we outperforms LaneSegNet by 4.0% mAP, 3.1% $\text{TOP}_{lsls}$ and 4.1% $\text{Acc}_b$, and exceeds Topo2Seq by 3.0% mAP. This shows the effectiveness of our streaming design for lane segment. Compared with temporal detection methods, we achieve a remarkable improvement of 10.6% mAP, 11.9% $\text{TOP}_{lsls}$ and 10.6% $\text{Acc}_b$. They exhibit limitations in detecting more fragmented lane segments, as they fail to account for multiple attributes and PE design.

**Results on Centerline Perception** The results of centerline perception are shown in Tab. 3. Compared with TopoFormer, our method achieve superior OLS (**44.4** v.s. 42.1) and topology reasoning capacity (**28.8** v.s. 24.7). This is attributed to the integration of auxiliary denoising training, PE design, and multi-attribute constraints in temporal detection.

### 4.4 MODEL ANALYSIS

**Ablation Studies for Streaming Attribute Constraints**. The results are shown in Tab. 4a. The baseline implementation, corresponding to the first row, incorporates the DBPE module into the streaming framework while excluding both the streaming attribute constraints and lane segment denoising components. Introducing class constraint in streaming can achieve a considerable improvement in lane boundary classification. Subsequently, the progressive integration of mask and coordinate constraints leads to enhanced detection capability and improved topology reasoning per-

**Table 4:** Ablation study results on losses.

| $\mathcal{L}_{tranloss}$ | $\mathcal{L}_{cls}^{Stream}$ | $\mathcal{L}_{mask}^{Stream}$ | $\mathcal{L}_{coord}^{Stream}$ | mAP | TOP$_{lsls}$ | Acc$_b$ |
|---|---|---|---|---|---|---|
| | | | | 33.8 | 26.1 | 47.8 |
| ✓ | | | | 34.7 | 26.3 | 49.0 |
| | ✓ | | | 34.0 | 26.5 | 49.2 |
| | ✓ | ✓ | | 35.1 | 27.0 | 49.2 |
| | ✓ | ✓ | ✓ | **35.6** | **27.8** | **49.5** |

| $\mathcal{L}_{dqd}$ | $\mathcal{L}_{cls}^{DN}$ | $\mathcal{L}_{topo}^{DN}$ | $\mathcal{L}_{coord}^{DN}$ | mAP | TOP$_{lsls}$ | Acc$_b$ |
|---|---|---|---|---|---|---|
| | | | | 33.8 | 26.1 | 47.8 |
| ✓ | | | | 35.2 | 27.2 | 49.6 |
| | ✓ | | | 35.1 | 27.0 | 49.5 |
| | ✓ | ✓ | | 35.7 | 27.9 | 49.7 |
| | ✓ | ✓ | ✓ | **36.3** | **28.0** | **49.8** |

**(a)** Ablation studies on streaming attribute constraints and transformation loss (Yuan et al., 2024).

**(b)** Ablation studies on lane segment denoising and dynamic query denoising (Wang et al., 2024b).

formance. The transformation loss in StreamMapNet (Yuan et al., 2024) only propagates and constrains the coordinates of line during temporal propagation. In contrast, our stream attribute constraints transfer and enforce the consistency of centerlines, boundary coordinates, typesand semantic masks. This plays a crucial role in the temporal detection of lane segments with multiple attributes. Consequently, compared to the transformation loss, our stream attribute constraints achieve an improvement of 0.9% in mAP and 1.5% in TOP$_{lsls}$.

**Ablation Studies for Lane Segment Denoising**. The results are presented in Tab. 4b. The baseline implementation remains consistent with Tab. 4a. Incorporating class denoising for content information in queries enhances performance in both detection and lane boundary classification, highlighting the importance of content learning in perception. Additionally,

**Table 5:** Ablation study on different modules in our method.

| Method | mAP | AP$_{ls}$ | AP$_{ped}$ | TOP$_{lsls}$ | Acc$_b$ |
|---|---|---|---|---|---|
| BL(w. Static PE) | 32.6 | 32.3 | 32.9 | 25.4 | 45.9 |
| +DBPE | 33.5 | 32.6 | 34.3 | 26.1 | 47.2 |
| +DBPE+LSDN | 34.5 | 32.9 | 36.1 | 26.7 | 47.6 |
| +Stream+Static PE | 33.8 | 32.8 | 34.8 | 26.3 | 47.7 |
| +Stream w/o PE | 33.5 | 32.0 | 34.9 | 27.1 | 48.3 |
| +Stream+DBPE | 35.6 | 34.7 | 36.5 | 27.8 | 49.5 |
| +Stream+DBPE+LSDN w/o IDTrack | 34.4 | 33.6 | 36.2 | 28.0 | 49.5 |
| +Stream+CPE((Liu et al., 2024))+LSDN | 34.5 | 34.0 | 35.0 | 28.1 | 49.7 |
| +Stream+DBPE+LSDN | **36.6** | **35.0** | **38.1** | **28.5** | **50.0** |

topology denoising enhances the robustness of topological reasoning against coordinate noise, while coordinate denoising boosts detection performance. Existing denoising methods only denoise center coordinates and categories. Our method extends this by also denoising additional attributes like boundary lines and topological relations for lane segments. This yields a performance gain of 1.1% mAP and 0.8% TOP$_{lsls}$ compared to dynamic query denoising.

**Module Ablations**. The first row in Tab. 5 presents our baseline (BL) model, LaneSegNet. LaneSegNet injects static PE into the queries. With the introduction of dynamic lane boundary PE (DBPE), the model exhibits a slight improvement. Further enhancement is achieved by incorporating lane segment denoising (LSDN). These results demonstrate that the incorporation of dynamic lane boundary PE and lane segment denoising effectively improves the overall performance of the per-frame detection model. Comparing row 6 with row 2, when the baseline model is adapted to the streaming paradigm and supervised with streaming attribute constraints, considerable improvements are observed (**35.6**% v.s. 33.5% in mAP, **27.8**% v.s. 26.1% in TOP$_{lsls}$, and **49.5**% v.s. 47.2% in Acc$_b$). However, substituting DBPE with either initial static PE or no PE at all adversely impacts performance, particularly resulting in a 2% reduction in mAP. Compared with centerline PE (CPE) (Liu et al., 2024), our DBPE achieves an improvement of 2.1 mAP. This demonstrates that injecting positional encoding via boundary points is particularly beneficial for recognizing areal lane segments, especially in detecting pedestrian crossings. Finally, the addition of denoising leads to optimal performance. However, as demonstrated in the row 7, the model exhibits a significant decline in detection performance when the unique IDs of positive instances are not tracked within streaming attribute constraints to ensure lossless supervision.

### 4.5 Qualitative Results

As shown in Fig. 5, 6 and 7, TopoStreamer is capable of predicting a complete road network with clearly lane boundaries, accurate topology connections, and temporal consistency. Additional qualitative results are provided in the appendix.

### 5 Conclusion

In this paper, we propose TopoStreamer, a temporal lane segment perception model for lane topology reasoning. Specifically, we incorporate three novel modules into an end-to-end network. The

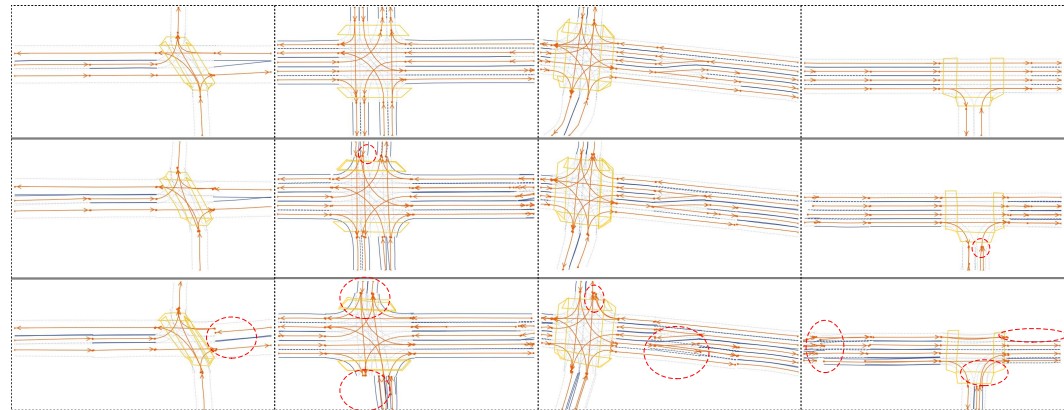

**Figure 6:** Qualitative results under different road structures. The images from top to bottom correspond to GT, our TopoStreamer, and LaneSegNet(Li et al., 2023b). For better viewing, zoom in on the image.

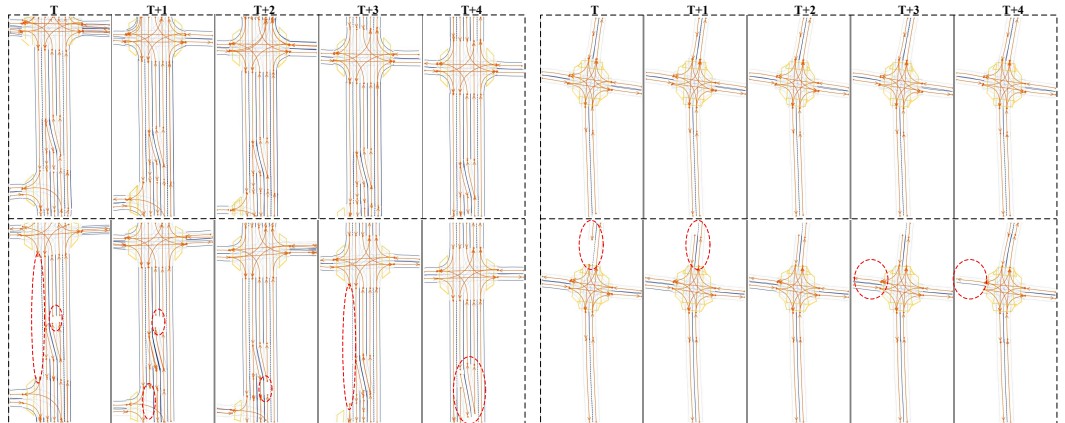

(a) Results when the ego vehicle is moving.      (b) Results when the ego vehicle is stationary.

**Figure 7:** Visualization of topology predictions across consecutive 5 frames. The results of TopoStreamer are shown on the top, and the results of LaneSegNet (Li et al., 2023b) are shown on the bottom. For better viewing, zoom in on the images. This demonstrates TopoStreamer's capability for accurate detection with temporal consistency.

streaming attribute constraints ensure the temporal consistency of both centerline and boundary coordinates, along with their classifications. Meanwhile, dynamic lane boundary positional encoding enhances the up-to-date positional information learning in queries, and lane segment denoising facilitates the learning of diverse patterns within lane segments. Furthermore, we evaluate the accuracy of existing models on our newly proposed lane boundary classification metric, which serves as a crucial measure of lane-changing scenarios in autonomous driving. Experimental results on the OpenLane-V2 dataset demonstrate the strong performance of our model and the effectiveness of our proposed designs.

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

# A APPENDIX

## A.1 STREAMING ATTRIBUTE CONSTRAINTS.

**Stream memory.** To facilitate temporal fusion, we introduce several memory modules, including stream query memory, stream BEV memory, and stream reference point memory, which store the predictions from the preceding frame. Specifically, for the detection results in the frame at timestamp T-1, we rank the queries based on their classification confidence scores and select the top-K queries for temporal propagation. In our implementation, K is set to 66, corresponding to the top 30% of queries. These selected queries, along with their associated centerline reference points and boundary reference points, are stored in their respective memory banks. Additionally, the BEV feature of the scene at T-1 is also stored in a memory. When processing the frame at timestamp T, the stored queries and their corresponding centerline reference points and boundary reference points are retrieved and undergo the following transformation by using transformation matrix:

$$
\begin{aligned}
\mathbf{Q_t^S} &= \text{MLP}(\text{Concat}(\mathbf{Q_{t-1}}, \Psi)) + \mathbf{Q_{t-1}} \\
\mathbf{R_B^S} &= \text{Norm}(\text{Concat}(\Psi \cdot \tilde{\mathbf{L}}_{t-1}^c, \Psi \cdot \tilde{\mathbf{L}}_{t-1}^l, \Psi \cdot \tilde{\mathbf{L}}_{t-1}^r))
\end{aligned}
\tag{1}
$$

where $\text{Norm}(\cdot)$ denotes the normalization operation applied to the coordinates.

The stored BEV feature of the scene at T-1 is also retrieved from memory, transformed, and then fused with the BEV feature extracted from the current frame (T):

$$
\tilde{\mathbf{F}}_{bev}^t = \text{GRU}(\text{Warp}(\mathbf{F}_{bev}^{t-1}, \Psi), \mathbf{F}_{bev}^t)
\tag{2}
$$

By doing so, our stream memories effectively integrates historical information, thereby enhancing the detection performance for the current frame.

We employ MLPs to predict lane segment coordinate, lane segment class, boundary class and BEV mask from stream queries $\mathbf{Q_t^S}$:

$$
\begin{aligned}
\tilde{\mathbf{L}}_t^c &= \text{MLP}_{reg}(\mathbf{Q_t^S}) + \text{InSigmod}(\mathbf{R_C^S}) \\
\tilde{\mathbf{L}}_t^c &= \text{Denorm}(\text{sigmoid}(\tilde{\mathbf{L}}_t^c)) \\
offset &= \text{MLP}_{offset}(\mathbf{Q_t^S}) \\
\tilde{\mathbf{L}}_t^l &= \tilde{\mathbf{L}}_t^c + offset, \tilde{\mathbf{L}}_t^r = \tilde{\mathbf{L}}_t^c - offset \\
\tilde{\mathbf{L}}_t &= \text{Concat}(\tilde{\mathbf{L}}_t^c, \tilde{\mathbf{L}}_t^l, \tilde{\mathbf{L}}_t^r) \\
\tilde{C}_t &= \text{MLP}_{cls}(\mathbf{Q_t^S}) \\
\tilde{T}_t &= \text{MLP}_{bcls}(\mathbf{Q_t^S}) \\
\tilde{\mathbf{M}}_t &= \text{Sigmoid}(\text{MLP}_{mask}(\mathbf{Q_t^S}) \otimes \tilde{\mathbf{F}}_{bev}^t)
\end{aligned}
\tag{3}
$$

where InSigmod refers to the inverse sigmoid function, while Denorm stands for denormalize. Then, the streaming attribute constraints are represented as:

$$
\begin{aligned}
\mathcal{L}_{coord}^{Stream} &= \mathcal{L}_{L1}(\tilde{\mathbf{L}}_t, \mathbf{L}_t) \\
\mathcal{L}_{cls}^{Stream} &= \kappa_1 \mathcal{L}_{Focal}(\tilde{C}_t, C_t) + \kappa_2 \mathcal{L}_{CE}(\tilde{T}_t, T_t) \\
\mathcal{L}_{mask}^{Stream} &= \kappa_3 \mathcal{L}_{CE}(\tilde{\mathbf{M}}_t, \mathbf{M}_t) + \kappa_4 \mathcal{L}_{Dice}(\tilde{\mathbf{M}}_t, \mathbf{M}_t) \\
\mathcal{L}_{Stream} &= \kappa_5 \mathcal{L}_{coord}^{Stream} + \kappa_6 \mathcal{L}_{cls}^{Stream} + \kappa_7 \mathcal{L}_{mask}^{Stream}
\end{aligned}
\tag{4}
$$

where the values of $\kappa_1$, $\kappa_2$, $\kappa_3$, $\kappa_4$, $\kappa_5$, $\kappa_6$, and $\kappa_7$ are 1.5, 0.01, 1.0, 1.0, 0.025, 1.0 and 3.0. $\mathbf{L}_t$, $T_t$, $C_t$ and $\mathbf{M}_t$ are GT annotations transformed from T-1 frame to T frame.

## A.2 LANE SEGMENT DENOISING

Lane segment denoising applies controlled noise to annotations and then removes it, thereby improving the model's capability to learn the diverse patterns present in lane segments. In the lane

segment denoising, we predict position, classification and adjacency matrix from denoising (DN) queries $\mathbf{Q^D}$ as follows:

$$\tilde{\mathbf{L}}^c = \text{MLP}_{reg}(\mathbf{Q^D}) + \text{InSigmod}(\mathbf{R_C^D})$$
$$\tilde{\mathbf{L}}^c = \text{Denorm}(\text{sigmoid}(\tilde{\mathbf{L}}^c))$$
$$offset = \text{MLP}_{offset}(\mathbf{Q^D})$$
$$\tilde{\mathbf{L}}^l = \tilde{L}_t^c + offset, \tilde{\mathbf{L}}^r = \tilde{\mathbf{L}}_t^c - offset$$
$$\tilde{\mathbf{L}} = \text{Concat}(\tilde{\mathbf{L}}_t^c, \tilde{\mathbf{L}}^l, \tilde{\mathbf{L}}_t^r) \qquad (5)$$
$$\tilde{C} = \text{MLP}_{cls}(\mathbf{Q^D})$$
$$\tilde{T} = \text{MLP}_{bcls}(\mathbf{Q^D})$$
$$\mathbf{Q^{D'}} = \text{MLP}_{pre}(\mathbf{Q^D}), \mathbf{Q^{D''}} = \text{MLP}_{suc}(\mathbf{Q^D})$$
$$\tilde{\mathbf{A}} = \text{Sigmoid}(\text{MLP}_{topo}(\text{Concat}(\mathbf{Q^{D'}}, \mathbf{Q^{D''}})))$$

Then, the denoising loss function is defined as:

$$\mathcal{L}_{coord}^{DN} = \mathcal{L}_{L1}(\tilde{\mathbf{L}}, \mathbf{L})$$
$$\mathcal{L}_{cls}^{DN} = \beta_1 \mathcal{L}_{Focal}(\tilde{C}, C) + \beta_2 \mathcal{L}_{CE}(\tilde{T}, T)$$
$$\mathcal{L}_{Topo}^{DN} = \mathcal{L}_{Focal}(\tilde{\mathbf{A}}, \mathbf{A}) \qquad (6)$$
$$\mathcal{L}_{denoise} = \lambda_1 \mathcal{L}_{coord}^{DN} + \lambda_2 \mathcal{L}_{cls}^{DN} + \lambda_3 \mathcal{L}_{topo}^{DN}$$

where the hyperparameters are defined as: $\beta_1 = 1.5$, $\beta_2 = 0.01$, $\lambda_1 = 0.025$, $\lambda_2 = 1.0$ and $\lambda_3 = 5.0$. Some examples of lane segment denoising are shown in Fig. 2. For better visualization, we only display the denoising results of the centerlines. It can be observed that the added noise disrupts the connectivity of the road network. Through the denoising process, the original positions and connectivity relationships are effectively restored. This enhances the model's ability to predict both the positional and connectivity topology of lane segments.

### A.3 TOTAL LOSS FUNCTION

The overall loss function in TopoSteamer is defined as:

$$\mathcal{L} = \alpha_1 \mathcal{L}_{ls} + \alpha_2 \mathcal{L}_{stream} + \alpha_3 \mathcal{L}_{denoise} \qquad (7)$$

where $\alpha_1 = 1.0$, $\alpha_1 = 0.3$ and $\alpha_1 = 1.0$, respectively. The lane segment loss is defined as:

$$\mathcal{L}_{ls} = \omega_1 \mathcal{L}_{vec} + \omega_2 \mathcal{L}_{seg} + \omega_3 \mathcal{L}_{cls} + \omega_4 \mathcal{L}_{type} + \omega_5 \mathcal{L}_{topo} \qquad (8)$$

where $\mathcal{L}_{seg} = \omega_6 \mathcal{L}_{ce} + \omega_7 \mathcal{L}_{dice}$ consists of a Cross-Entropy loss and a Dice loss used to supervise the BEV semantic mask., and the hyperparameters are defined as: $\omega_1 = 0.025$, $\omega_2 = 3.0$, $\omega_3 = 1.5$, $\omega_4 = 0.01$, $\omega_5 = 5.0$, $\omega_6 = 1.0$ and $\omega_7 = 1.0$. $\mathcal{L}_{vec}$ is the L1 loss computed between the predicted vectorized lanes and the ground truth lanes. The classification losses $\mathcal{L}_{cls}$ and $\mathcal{L}_{type}$ are used for lane segment classification. $\mathcal{L}_{topo}$ is a focal loss applied to supervise the topological relationship prediction. It is worth noting that the weighting strategies for the losses related to different lane segment attributes in both streaming attribute constraints and lane segment denoising are consistent with the loss configurations in LaneSegNet.

### A.4 STREAMING TRAINING

We adopt the streaming training strategy for temporal fusion. For each training sequence, we randomly divide it into 2 splits at the start of each training epoch to foster more diverse data sequences. During inference, we use the entire sequences. Suppose a batch contains N samples, each from a different scene, read in chronological order. Temporal fusion is performed by determining whether the current data and the previously read data belong to the same scene.

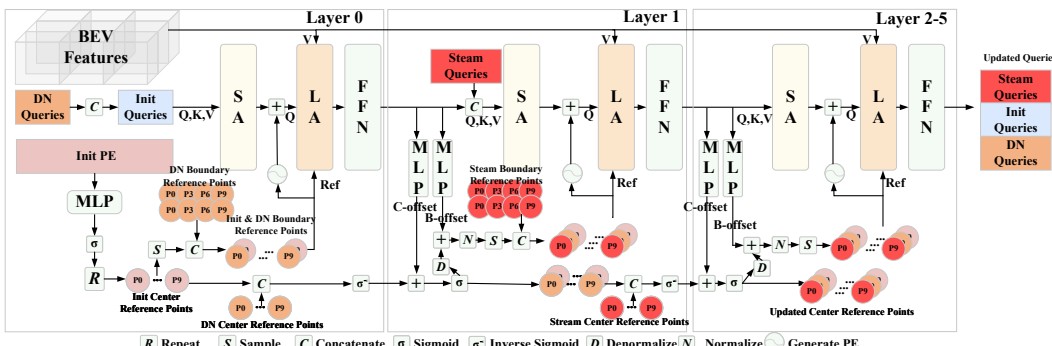

**Figure 1:** The detailed architecture of decoder.

| Method | Venue | Epochs | Temporal | OLS ↑ | DET$_l$ ↑ | TOP$_{ll}$ ↑ |
|---|---|---|---|---|---|---|
| VectorMapNet (Liu et al., 2023) | ICML23 | 24 | No | - | 3.5 | - |
| STSU (Can et al., 2021) | ICCV21 | 24 | No | - | 8.2 | - |
| MapTR (Liao et al., 2022) | ICLR23 | 24 | No | - | 15.2 | - |
| TopoNet (Li et al., 2023a) | Arxiv23 | 24 | No | 25.1 | 24.3 | 6.7 |
| TopoMLP (Wu et al., 2023) | ICLR24 | 24 | No | 36.2 | 26.6 | 19.8 |
| LaneSegNet (Li et al., 2023b) | ICLR24 | 24 | No | 38.7 | 27.5 | 24.9 |
| TopoLogic (Fu et al., 2025a) | NIPS24 | 24 | No | 36.2 | 25.9 | 21.6 |
| StreamMapNet (Yuan et al., 2024) | WACV24 | 24 | Yes | 26.7 | 18.9 | 11.9 |
| SQD-MapNet (Wang et al., 2024b) | ECCV24 | 24 | Yes | 29.1 | 21.9 | 13.3 |
| **TopoStreamer (ours)** | - | 24 | Yes | **42.6** | **30.9** | **29.4** |

**Table 1:** Comparison with the state-of-the-arts on OpenLane-V2 subsetB on centerline perception.

## A.5 METRIC FOR LANE BOUNDARY CLASSIFICATION

Previous approach (Li et al., 2023b) classify lane boundaries as dashed, solid, or non-visible, but they don't measure how accurate these predictions are. This accuracy is crucial for self-driving cars when making lane-change decisions. To solve this, we introduce a new metric to to evaluate lane boundary classification accuracy. We follow the design of Top$_{lsls}$ metric (Wang et al., 2024a). We first build a projection between predictions and ground truth to preserve true positive instances, according to Fréchet distance. Then, we evaluate the accuracy of the left and right boundary types by comparing the predicted types with the GT types.

## A.6 DECODER ARCHITECTURE

The detailed architecture of the decoder is shown in Fig. 1. In the first layer of the decoder, we utilize identical initialization (Li et al., 2023b) to produce the initial centerline reference points and boundary reference points from the initial position embedding. The initial queries, combined with DN queries, are fed into the self-attention (SA) and then augmented with the position embedding generated from the initial and DN boundary reference points. The sampling of boundary reference points is based on the heads-to-regions sampling method (Li et al., 2023b). Specifically, sampling is performed at symmetric offset positions on both sides of the centerline reference points. The position embedding is obtained by applying sinusoidal encoding to the coordinates of the reference point. These queries interact with BEV features through lane attention (Li et al., 2023b), employing a heads-to-regions sampling mechanism guided by the boundary reference points. At the outset of layer 1, we predict an updated offset to refine the initial and DN center reference points, along with a boundary offset to determine the boundary reference points by applying it to the center reference points. Meanwhile, in this layer, stream query, stream center reference points and stream boundary reference points are employed to replace the lowest confidence N-k queries and their reference points. Here, N represents the predefined total number of queries, set at 200, while K denotes the number of stream queries, which is 66, accounting for 30% of the total queries. The same updating procedure as in layer 0 is then applied to these queries. The updating process remains consistent and regular across layers 2 to 5.

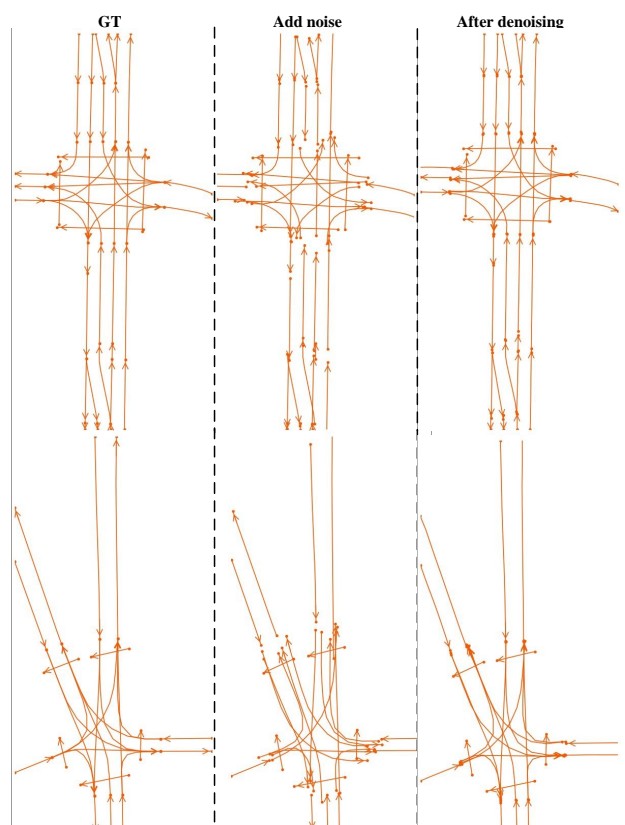

**Figure 2:** Qualitative results for lane segment denoising.

(a) Different numbers of stream queries.

| Top-K | mAP | TOP$_{lsls}$ |
|---|---|---|
| 10% | 34.8 | 27.9 |
| **30%** | **36.6** | **28.5** |
| 50% | 34.3 | 27.3 |
| 75% | 32.9 | 25.1 |

(b) Different numbers of DN queries.

| Number | mAP | TOP$_{lsls}$ |
|---|---|---|
| 120 | 35.0 | 27.4 |
| **240** | **36.6** | **28.5** |
| 360 | 35.1 | 27.7 |

**Table 2:** Ablation study results.

### A.7 EXPERIMENT

We provide comparative experiments on the OpenLane-V2 subset-B benchmark. In fact, this benchmark do not contain lane segment annotations with only centerline annotations. We generate pseudo-labels by augmenting the lane centerlines with a standardized lane width of 4 meters. The results are shown in Tab. 1. We outperforms LaneSegNet by 3.9% OLS, 3.4% DET$_l$, and 4.5 % TOP$_{ll}$.

We present additional experiments focusing on the selection of the number of stream queries and DN queries on subset-A. The results of the ablation study investigating the impact of varying numbers of stream queries are presented in Tab. 2a. Optimal performance is attained when 30% of the queries from the preceding frame are selected for temporal propagation. The results of the ablation study about the number of DN queries are shown in Tab. 2b. Setting the number of DN queries to 240 yields the optimal performance.

### A.8 DEMO

See the supplementary material vis.gif file for details.

| Method | FLOPs | Param |
|---|---|---|
| LaneSegNet | 639.1G | 30.9M |
| TopoStreamer | 652.1G | 46.2M |

**Table 3:** Comparison of computational complexity.

### A.9  TRAINING AND TESTING TIMES

The training and inference times are related to the model and quantity of the GPUs used. Our setup utilizes 4 V100 GPUs, with a training time of approximately 28 hours and an testing time of about 40 minutes. Increasing the number of GPUs can accelerate the process, and switching to 4 A100 GPUs can reduce the training time to 20 hours.

### A.10  COMPUTATIONAL COMPLEXITY

The computational complexity comparison against the our baseline model LaneSegNet is shown in Table 3. We use the FLOPs (Floating Point Operations) and the number of parameters (Params) of the model during inference to represent the computational cost. The additional parameters in TopoStreamer, compared to LaneSegNet, are introduced by the temporal fusion and denoising modules. Furthermore, TopoStreamer only results in a marginal increase in FLOPs. The slight drop in FPS of our method is a direct result of the computational overhead introduced by our temporal propagation framework. This modest computational overhead is justified by a performance gain of 4.0% in mAP and 3.1% in TOP$_{lsls}$

### A.11  LIMITATION AND FUTURE WORK

Current lane topology reasoning methods are affected by the long-tail problem, exhibiting limited detection confidence in regions with excessive curvature or indistinct lane markings. Therefore, we plan to explore the use of Vision-Language Models (VLMs) to address this issue. By leveraging VLMs to interpret road structures and generate Chains of Thought (CoT), we aim to provide prior knowledge for lane topology reasoning and enhance interpretability. Additionally, we will investigate integrating road topology with end-to-end autonomous driving systems, using lane topology to constrain vehicle trajectory planning and improve safety.

### A.12  USE OF LLM

In this paper, large language model is used only for writing enhancement purposes.

