# OpenReview forum: "TopoStreamer: Temporal Lane Segment Topology Reasoning in Autonomous Driving"
_ICLR.cc/2026/Conference — Submitted to ICLR 2026_

### Official Review · Reviewer_h3uG · 2025-10-29

**Soundness:** 1
**Presentation:** 1
**Contribution:** 2
**Rating:** 2
**Confidence:** 4

**Summary:**

1. This paper designs a comprehensive loss function that imposes consistency constraints on the temporal propagation of lane centerline and boundary line coordinates as well as their classification information, ensuring the stability of multidimensional attributes.
2 .DBPE: In the layer-by-layer forward propagation of the decoder, it resolves the conflict between the static nature of traditional positional encoding and the dynamic updating of reference points, significantly improving localization accuracy.

**Strengths:**

1. Introducing temporaling modeling into the lane segment perception task is valuable for stable map construction.
2. Experimental results show that on the OpenLane-V2 dataset, TopoStreamer achieves a 3.0% mAP improvement in lane segment perception and a 1.7% OLS improvement in centerline perception compared to the previous state-of-the-art (SOTA)

**Weaknesses:**

1. Many parts of this work claimed as core contributions have similar spirits to other autonomous driving works like StreamMapNet, and etc. Overall, this paper mainly solves several problems when such temporal modeling is applied to this specific lane segmentation prediction task.

2. Streaming Attribute Constraints seems to be an auxiliary loss that adds direct supervision to the results of stream queries. Compared to the previous transformation loss, it seems to have a similar effect but is implemented differently. It seems to be a trick.

3. The position of the ) in equation 4 seems to be wrong; inside LA it should be query, F_bev, and R.

4. The symbols in the paper are difficult to understand. Some symbols in the paper, from lines 167-181, should be clearly explained. Some symbol definitions are inconsistent; for example, some symbols are missing \mathbf{}.

5. The details of stream memory are not well described.

6. Some of the proposed methods, such as Streaming Attribute Constraints, DBPE, seem to be optimization tricks based on SDQ-MapNet, MapQR, and LaneSegNet. Could you emphasize the main motivation and contribution of this paper compared to previous methods?

6. There are too many hyperparameters related to the loss in the method. Although the ablation experiments and supplementary materials demonstrate effectiveness, it is challenging to fully ensure that each loss is valid and the hyperparameters are appropriately set.

**Questions:**

The questions have been included in the points of weakness.

---

> ### Author Response · Authors · 2025-11-14
> **Response to Reviewer h3uG (part I)**
>
> ### Author Response
>
> We sincerely thank the reviewers for their recognition of our work. Based on the valuable feedback provided, we have made substantial revisions to the manuscript. We kindly invite the reviewers to refer to our updated version when reviewing our responses. Below, we provide point-by-point answers to the raised questions:
>
> ---
>
> ### 1. Comparison with Existing Methods
>
> We acknowledge that existing methods have inspired our work. However, compared with the problems those methods focus on, **lane segments** possess multiple attributes such as left and right boundary coordinates and types, centerline position and type, and topological relationships. Existing methods are not directly applicable to lane segment topology reasoning; therefore, we designed a **temporal propagation framework specifically tailored for lane segments**.
>
> **Table4(a):** *Ablation studies on streaming attribute constraints and transformation loss[1]* .
>
> | $\mathcal{L}_{tranloss}$ | $\mathcal{L}_{cls}^{Stream}$ | $\mathcal{L}_{mask}^{Stream}$ | $\mathcal{L}_{coord}^{Stream}$ | mAP | TOP$_{lsls}$ | Acc$_b$ |
> |:-------------------------:|:----------------------------:|:-----------------------------:|:-------------------------------:|:----:|:-------------:|:--------:|
> |                           |                              |                               |                                 | 33.8 | 26.1 | 47.8 |
> | ✅                        |                              |                               |                                 | 34.7 | 26.3 | 49.0 |
> |                           | ✅                            |                               |                                 | 34.0 | 26.5 | 49.2 |
> |                           | ✅                            | ✅                             |                                 | 35.1 | 27.0 | 49.2 |
> |                           | ✅                            | ✅                             | ✅                               | **35.6** | **27.8** | **49.5** |
>
>
> **Table5:** *Ablation study on different modules in our method.*
>
> | **Method** | **mAP** | **AP$_{ls}$** | **AP$_{ped}$** | **TOP$_{lsls}$** | **Acc$_b$** |
> |:------------|:------:|:--------------:|:---------------:|:----------------:|:------------:|
> | BL (w. Static PE) | 32.6 | 32.3 | 32.9 | 25.4 | 45.9 |
> | +DBPE | 33.5 | 32.6 | 34.3 | 26.1 | 47.2 |
> | +DBPE + LSDN | 34.5 | 32.9 | 36.1 | 26.7 | 47.6 |
> | +Stream + Static PE[3] | 33.8 | 32.8 | 34.8 | 26.3 | 47.7 |
> | +Stream w/o PE[1] | 33.5 | 32.0 | 34.9 | 27.1 | 48.3 |
> | +Stream + DBPE | 35.6 | 34.7 | 36.5 | 27.8 | 49.5 |
> | +Stream + DBPE + LSDN w/o IDTrack | 34.4 | 33.6 | 36.2 | 28.0 | 49.5 |
> | +Stream + CPE[2] + LSDN | 34.5 | 34.0 | 35.0 | 28.1 | 49.7 |
> | **+Stream + DBPE + LSDN** | **36.6** | **35.0** | **38.1** | **28.5** | **50.0** |
>
>
> Compared with **StreamMapNet[1]**, it only propagates and constrains the coordinates of polylines during temporal propagation. In contrast, our **streaming attribute constraints** transfer and enforce the consistency of **centerlines, boundary coordinates, types, and semantic masks**. This mechanism plays a crucial role in the temporal detection of lane segments with multiple attributes. We provide an experimental comparison in **Table 4(a)** of the revised paper with StreamMapNet; consequently, our streaming attribute constraints achieve an **improvement of 0.9% in mAP and 1.5% in TOP$_{lsls}$**.
>
> In addition, we introduce an extra **Lossless Streaming Supervision**. As shown in **Table 5**, **+Stream + DBPE + LSDN** outperforms **+Stream + DBPE + LSDN w/o IDTrack** by **2.2% in mAP**.
> To clarify the distinctions of our method, relevant explanations have been added in the revised manuscript, **highlighted in blue at lines 216–220 and 442–447**, and the comparative results are presented in **Table 4(a)** of the revised paper.

---

> ### Author Response · Authors · 2025-11-14
> **Response to Reviewer h3uG (part II)**
>
> Compared with **MapQR[2]**, MapQR primarily injects PE for **centerlines**, which may reduce attention to positions along the boundary lines and conflict with our goal of predicting multiple attributes for the **entire lane segment area**. Therefore, considering the importance of **boundary point sampling** for lane segment area detection, we innovatively refine and encode the positions of boundary points at each layer to inject real-time updated positional information. In **Table 5** of the revised paper, we add additional experiments comparing our approach with the centerline-based PE (CPE) used in MapQR.
> **+Stream + DBPE + LSDN** achieves a **2.1% gain in mAP** compared to **+Stream + CPE + LSDN**, demonstrating that injecting positional encoding via boundary points is particularly beneficial for recognizing areal lane segments, especially in detecting pedestrian crossings. Relevant explanations have been added and **highlighted in red at lines 260–264 and 470–472**.
>
> **Table4(b):** *Ablation studies on lane segment denoising and dynamic query denoising[4]*
>
> | $\mathcal{L}_{dqd}$ | $\mathcal{L}_{cls}^{DN}$ | $\mathcal{L}_{topo}^{DN}$ | $\mathcal{L}_{coord}^{DN}$ | **mAP** | **TOP$_{lsls}$** | **Acc$_b$** |
> |:--------------------:|:------------------------:|:--------------------------:|:----------------------------:|:------:|:----------------:|:------------:|
> |                      |                          |                            |                              | 33.8 | 26.1 | 47.8 |
> | ✅                   |                          |                            |                              | 35.2 | 27.2 | 49.6 |
> |                      | ✅                        |                            |                              | 35.1 | 27.0 | 49.5 |
> |                      | ✅                        | ✅                          |                              | 35.7 | 27.9 | 49.7 |
> |                      | ✅                        | ✅                          | ✅                            | **36.3** | **28.0** | **49.8** |
>
>
> Compared with **SQD-MapNet[4]**, we have made **key modifications to query denoising** so that it effectively aids lane segment localization and topology learning. Specifically, the **content embedding** integrates the categories of both boundary lines and centerlines for the noise–denoise process. Additionally, we introduce **extra denoising losses** for boundaries and topology.
> To further verify the effectiveness of our approach, we compare the denoising mechanisms in SQD-MapNet and our method, as shown in **Table 4(b)** of the revised manuscript. This comparison demonstrates a **performance gain of 1.1% in mAP** and **0.8% in TOP$_{lsls}$**. The corresponding results have been added to the **Table 4(b)**, with the new content **highlighted in green (lines 457–460)**.
>
> Compared with **LaneSegNet[3]**, our framework extends it to **temporal lane segment topology reasoning**.  With the module designed for **robust temporal learning of lane segment attributes**, **TopoStreamer** outperforms **LaneSegNet** by **4.0% in mAP** and **3.1% in TOP$_{lsls}$**.
>
> These results confirm that each module contributes to the overall improvement by comprehensively addressing the characteristics of lane segments. Seamlessly integrating all these modules into a unified end-to-end framework is both novel and effective. Moreover, we introduce lossless streaming supervision to strengthen temporal supervision and a metric to evaluate lane boundary classification accuracy, which further enhances the OpenLaneV2 benchmark.
>
>
> ---
>
> ### 2. Symbol Definitions
>
> Thank you for highlighting this issue. We have unified all symbol definitions and added a **nomenclature table (Table 1)** in the revised paper for clarity.
>
>
> **Table1:** *Meaning of the notations in TopoStreamer.*
>
> | **Notation** | **Meaning** |
> |:--------------|:------------|
> | $\mathbf{L}^c,\mathbf{L}^l,\mathbf{L}^r$ | Center, left boundary, and right boundary lines |
> | $\mathbf{F}_{bev}, \mathbf{F}_{pe}, \mathbf{F}_{content}$ | BEV feature, positional embedding, and content embedding |
> | $\mathbf{Q}$ | Queries |
> | $\mathbf{D}, \mathbf{S}, \mathbf{I}$ | Denoising (DN), stream, and initialized |
> | $t$ | Timestamp *T* |
> | $\mathbf{R_B}, \mathbf{R_C}$ | Boundary and center reference points |
> | $\mathit{C}, \mathit{T}$ | Lane segment class and boundary class |
> | $\mathbf{M}, \mathbf{A}$ | BEV semantic mask and adjacency matrix |
> | $\mathbf{P} = \{(x_i, y_i, z_i)\}$ | An ordered set of points that forms a lane |
> | $\Psi$ | Transformation matrix |

---

> ### Author Response · Authors · 2025-11-14
> **Response to Reviewer h3uG (part III)**
>
> ---
>
> ### 3. Stream Memory
>
> In response to the reviewer’s suggestion, we have provided additional details regarding the **stream memory** in **Section A.1** of the appendix (**highlighted in blue, lines 706–727**).
>
>
> ---
>
> ### 4. Correction of Equation (4)
>
> We appreciate the reviewer’s observation regarding the equation error. It has been corrected in the revised manuscript as follows:
>
> $$
> \begin{aligned}
> \textbf{F}_{pe} &= \{\text{PE}(\mathbf{P^B_i})\}, \\
> \tilde{\textbf{Q}} &= \text{MLP}\big(\text{LA}(\text{Duplicate}(\text{SA}(\textbf{Q})) + \textbf{F}_{pe},\; \tilde{\textbf{F}}_{bev},\; \mathbf{R_B})\big)
> \end{aligned}
> $$
>
> ---
>
> ### 5. Hyperparameter Selection for Loss Functions
>
> We follow the **baseline model LaneSegNet** for hyperparameter selection. The LaneSegNet loss is defined as:
>
> $$
> \mathcal{L}_{ls} = \omega_1 \mathcal{L}_{vec} + \omega_2 \mathcal{L}_{seg} + \omega_3 \mathcal{L}_{cls} + \omega_4
> \mathcal{L}_{type} + \omega_5 \mathcal{L}_{topo},
> $$
>
> where
>
> $$
> \mathcal{L}_{seg} = \omega_6\mathcal{L}_{ce} + \omega_7\mathcal{L}_{dice},
> $$
>
> consists of a Cross-Entropy loss and a Dice loss used to supervise the BEV semantic mask.
> The hyperparameters are:
> $\omega_1 = 0.025$, $\omega_2 = 3.0$, $\omega_3 = 1.5$, $\omega_4 = 0.01$, $\omega_5 = 5.0$, $\omega_6 = 1.0$, and $\omega_7 = 1.0$.
>
> In our **streaming attribute constraints** and **denoising losses**, we adopt similar **weighting strategies for various lane segment attribute constraints** as those used in **LaneSegNet**.
> Details of the loss functions can be found in **Sections A.1 and A.2** of the appendix.
> In addition, we have included references for the hyperparameter selection in **lines 800–802** (highlighted in green) of the appendix.
>
>
> ---
> [1] Streammapnet: Streaming mapping network for vectorized online hd map construction, WACV2024.
>
> [2] Leveraging enhanced queries of point sets for vectorized map construction, ECCV2024.
>
> [3] Lanesegnet: Map learning with lane segment perception for autonomous driving, ICLR2024.
>
> [4] Stream query denoising for vectorized hd-map construction, ECCV2024.

---

> ### Author Response · Authors · 2025-11-25
> **A Warm Reminder Regarding Our Previous Response**
>
> Dear Reviewer h3uG,
>
> I hope you are doing well. I am writing to kindly follow up on my earlier response. If there is anything further you would like me to clarify or provide, please feel free to let me know. Thank you again for your time and effort in reviewing our submission. I look forward to your reply.
>
> Best regards,
>
> Authors of Submission 360

---

> ### Author Response · Authors · 2025-11-28
> **A Warm Reminder Regarding Our Previous Response**
>
> Dear Reviewer h3uG,
>
> I hope you are doing well.
>
> I am writing to kindly follow up on our previous response. We hope our point-by-point rebuttal has adequately addressed the concerns you raised.
>
> Should any further clarification or additional information be needed, please do not hesitate to let me know.
>
> Thank you once again for your time and valuable effort in reviewing our submission . We look forward to hearing from you. Best regards,
>
> The Authors of Submission 360

---

### Official Review · Reviewer_sabF · 2025-11-01

**Soundness:** 2
**Presentation:** 2
**Contribution:** 1
**Rating:** 2
**Confidence:** 4

**Summary:**

The paper proposes TopoStreamer, a temporal perception framework for lane segment topology reasoning in autonomous driving. It introduces three components: streaming attribute constraints to enforce temporal consistency, dynamic lane boundary positional encoding (PE) for improving spatial localization, and lane segment denoising to handle diverse temporal patterns. The method is evaluated on the OpenLane-V2 benchmark and reports +3.0% mAP improvement in lane segment perception and +1.7% OLS in centerline perception over previous methods.

**Strengths:**

- The paper proposes a temporal perception model that explicitly addresses temporal consistency and positional embedding issues in lane topology reasoning.
- The paper includes comparisons with multiple strong baselines and extensive quantitative ablations, demonstrating solid engineering effort. The proposed method achieves state-of-the-art (SOTA) results in both lane segment and centerline perception tasks.

**Weaknesses:**

- The main contributions — temporal propagation, positional encoding refinement, and query denoising — are all direct extensions of existing methods.The proposed modules are minor architectural tweaks rather than conceptual advances.
- The paper lacks a detailed analysis of computational complexity (e.g., FLOPs, parameter count) and its comparison with baseline methods.

**Questions:**

- In Table 1, why does TOPlsls of TopoStreamer remain lower than TopoLogic, despite your method being temporal and supposedly more consistent? Would integrating TopoLogic’s topology reasoning potentially improve your model’s results?
- TopoStreamer achieves only higher FPS than TopoNet, and is slower than other recent baselines. Does this imply that the proposed model trades off speed for accuracy?

---

> ### Author Response · Authors · 2025-11-14
> **Response to Reviewer sabF (part I)**
>
> ### Author Response
>
> We sincerely thank the reviewers for their recognition of our work. Based on the valuable feedback provided, we have made substantial revisions to the manuscript. We kindly invite the reviewers to refer to our updated version when reviewing our responses. Below, we provide point-by-point answers to the raised questions:
>
> ---
>
> #### **1. Motivation of TopoStreamer**
>
> We acknowledge that existing methods have inspired our work. However, compared with the problems those methods focus on, **lane segments** possess multiple attributes such as left and right boundary coordinates and types, centerline position and type, and topological relationships. Existing methods are not directly applicable to lane segment topology reasoning; therefore, we designed a **temporal propagation framework specifically tailored for lane segments**.
>
> To achieve robust temporal propagation of multiple lane segment attributes, we propose **streaming attribute constraints**. The transformation loss in existing methods [1] only propagates and constrains the coordinates of polylines during temporal propagation. In contrast, our streaming attribute constraints transfer and enforce the consistency of **centerlines, boundary coordinates, types, and semantic masks**. This mechanism plays a crucial role in the temporal detection of lane segments with multiple attributes.
> **Table4(a):** *Ablation studies on streaming attribute constraints and transformation loss[1]* .
>
> | $\mathcal{L}_{tranloss}$ | $\mathcal{L}_{cls}^{Stream}$ | $\mathcal{L}_{mask}^{Stream}$ | $\mathcal{L}_{coord}^{Stream}$ | mAP | TOP$_{lsls}$ | Acc$_b$ |
> |:-------------------------:|:----------------------------:|:-----------------------------:|:-------------------------------:|:----:|:-------------:|:--------:|
> |                           |                              |                               |                                 | 33.8 | 26.1 | 47.8 |
> | ✅                        |                              |                               |                                 | 34.7 | 26.3 | 49.0 |
> |                           | ✅                            |                               |                                 | 34.0 | 26.5 | 49.2 |
> |                           | ✅                            | ✅                             |                                 | 35.1 | 27.0 | 49.2 |
> |                           | ✅                            | ✅                             | ✅                               | **35.6** | **27.8** | **49.5** |
>
> We provide an experimental comparison with the transformation loss; consequently, our streaming attribute constraints achieve an **improvement of 0.9% in mAP and 1.5% in TOP$_{lsls}$**. To clarify the distinctions of our method, relevant explanations have been added in the revised manuscript, **highlighted in blue at lines 216–220 and 442–447**, and the comparative results are presented in **Table 4(a)** of the revised paper. In addition, we introduce an extra **Lossless Streaming Supervision**.
> As shown in **Table 5**, **+Stream + DBPE + LSDN** outperforms **+Stream + DBPE + LSDN w/o IDTrack** by **2.2% in mAP**.
>
>
> **Table5:** *Ablation study on different modules in our method.*
>
> | **Method** | **mAP** | **AP$_{ls}$** | **AP$_{ped}$** | **TOP$_{lsls}$** | **Acc$_b$** |
> |:------------|:------:|:--------------:|:---------------:|:----------------:|:------------:|
> | BL (w. Static PE) | 32.6 | 32.3 | 32.9 | 25.4 | 45.9 |
> | +DBPE | 33.5 | 32.6 | 34.3 | 26.1 | 47.2 |
> | +DBPE + LSDN | 34.5 | 32.9 | 36.1 | 26.7 | 47.6 |
> | +Stream + Static PE[3] | 33.8 | 32.8 | 34.8 | 26.3 | 47.7 |
> | +Stream w/o PE[1] | 33.5 | 32.0 | 34.9 | 27.1 | 48.3 |
> | +Stream + DBPE | 35.6 | 34.7 | 36.5 | 27.8 | 49.5 |
> | +Stream + DBPE + LSDN w/o IDTrack | 34.4 | 33.6 | 36.2 | 28.0 | 49.5 |
> | +Stream + CPE[2] + LSDN | 34.5 | 34.0 | 35.0 | 28.1 | 49.7 |
> | **+Stream + DBPE + LSDN** | **36.6** | **35.0** | **38.1** | **28.5** | **50.0** |

---

> ### Author Response · Authors · 2025-11-14
> **Response to Reviewer sabF (part II)**
>
> Regarding **temporal positional encoding**, current temporal methods [1] do not incorporate positional encoding (PE), leading to inconsistent updates. Moreover, existing methods primarily inject PE for **centerlines** [2] or use **static PE** [3]. However, in lane segment recognition, this centerline-based PE injection may reduce attention to positions along the boundary lines, which conflicts with our goal of predicting multiple attributes for the **entire lane segment area**. Static PE injection also causes **inconsistencies between decoder layers**.
>
> Therefore, considering the importance of **boundary point sampling** for lane segment area detection, we innovatively refine and encode the positions of boundary points at each layer to inject real-time updated positional information. In **Table 5** of revised paper, we add additional experiments comparing our approach with static PE, no PE injection, and centerline PE injection. +Stream + DBPE achieves a **1.8% improvement in mAP** compared to +Stream w/o PE and +Stream + Static PE.
> Furthermore, +Stream + DBPE + LSDN achieves a **2.1% gain in mAP** compared to +Stream + centerline PE(CPE)+ LSDN. This demonstrates that injecting positional encoding via boundary points is particularly beneficial for recognizing areal lane segments, especially in detecting pedestrian crossings. Relevant explanations have been added and **highlighted in red at lines 260–264 and 470–472**.
>
> **Table4(b):** *Ablation studies on lane segment denoising and dynamic query denoising[4]*
>
> | $\mathcal{L}_{dqd}$ | $\mathcal{L}_{cls}^{DN}$ | $\mathcal{L}_{topo}^{DN}$ | $\mathcal{L}_{coord}^{DN}$ | **mAP** | **TOP$_{lsls}$** | **Acc$_b$** |
> |:--------------------:|:------------------------:|:--------------------------:|:----------------------------:|:------:|:----------------:|:------------:|
> |                      |                          |                            |                              | 33.8 | 26.1 | 47.8 |
> | ✅                   |                          |                            |                              | 35.2 | 27.2 | 49.6 |
> |                      | ✅                        |                            |                              | 35.1 | 27.0 | 49.5 |
> |                      | ✅                        | ✅                          |                              | 35.7 | 27.9 | 49.7 |
> |                      | ✅                        | ✅                          | ✅                            | **36.3** | **28.0** | **49.8** |
>
>
> We have also made **key modifications to query denoising** so that it effectively aids lane segment localization and topology learning. Specifically, the **content embedding** integrates the categories of both boundary lines and centerlines for the noise–denoise process. Additionally, we introduce **extra denoising losses** for boundaries and topology. To further verify the effectiveness of our approach, we compare it with a variant that only denoises centerline coordinates and classes[4], as shown in **Table 4(b)** of the revised manuscript.  This comparison demonstrates a **performance gain of 1.1% in mAP** and **0.8% in TOP$_{lsls}$**. The corresponding results have been added to the **Table 4(b)**, with the new content **highlighted in green (lines 457–460)**.
>
>
> These results confirm that each module contributes to the overall improvement by comprehensively addressing the characteristics of lane segments. Seamlessly integrating all these modules into a unified end-to-end framework is both **novel and effective**. Moreover, we introduce **lossless streaming supervision** to strengthen temporal supervision and a **metric to evaluate lane boundary classification accuracy**, which further enhances the **OpenLaneV2 benchmark**.
>
> ---
>
> #### **2. Analysis of computational complexity**
>
> **Table:** *Comparison of computational complexity.*
>
> | **Method** | **FLOPs** | **Params** |
> |:------------|:----------:|:-----------:|
> | LaneSegNet | 639.1G | 30.9M |
> | TopoStreamer | 652.1G | 46.2M |
>
> We thank the reviewer for this suggestion. We have added a **computational complexity comparison** against our baseline model **LaneSegNet[3]**. The additional parameters in **TopoStreamer**, compared with LaneSegNet, are introduced by the **temporal fusion** and **denoising modules**. Furthermore, **TopoStreamer only results in a marginal increase in FLOPs**, and this modest computational overhead is justified by a **performance gain of 4.0% in mAP and 3.1% in TOP$_{lsls}$**.
>
> The corresponding results have been added to the **appendix (Section A.10)** and **Table 3**, with the new content **highlighted in blue (lines 935–944)**.
>
> ---

---

> ### Author Response · Authors · 2025-11-14
> **Response to Reviewer sabF (part III)**
>
> #### **3. Integration of TopoStreamer and TopoLogic**
>
> We thank the reviewer for this insightful suggestion. Lane topology reasoning relies heavily on perception performance. Thus, our method focuses on boosting perception to enhance reasoning, instead of altering the structure of topology reasoning directly. **TopoLogic[5]** is an excellent work that introduces a more robust topological metric incorporating **lane geometric distances** and **semantic similarity of lane queries**, enhancing topological reasoning in cases of endpoint misalignment.
>
> **Table:** *Comparison between TopoStreamer and TopoStreamer with TopoLogic.*
>
> | **Method** | **mAP ↑** | **AP$_{ls}$ ↑** | **AP$_{ped}$ ↑** | **TOP$_{lsls}$ ↑** | **Acc$_b$ ↑** |
> |:-------------------------------|:---------:|:---------------:|:-----------------:|:----------------:|:-------------:|
> | **TopoStreamer** | **36.6** | 35.0 | **38.1** | 28.5 | 50.0 |
> | **TopoStreamer with TopoLogic** | 36.5 | **35.1** | 37.8 | **30.1** | **50.2** |
>
>
> We have **integrated the topological reasoning module from TopoLogic into TopoStreamer**, which has led to a **1.6% gain in TOP$_{lsls}$**. The corresponding results have been updated in **Table 2** of the revised manuscript.
>
> ---
>
> #### **4. Trade-off between speed and accuracy**
>
> The slightly lower FPS of our model is a direct result of the computational overhead introduced by the **temporal propagation framework**.  This trade-off allows our model to achieve **better overall performance**. We appreciate the reviewer’s observation and have discussed this trade-off in **Section A.10** of the appendix (**highlighted in blue, lines 934–943**).
>
> ---
> [1] Streammapnet: Streaming mapping network for vectorized online hd map construction, WACV2024.
>
> [2] Leveraging enhanced queries of point sets for vectorized map construction, ECCV2024.
>
> [3] Lanesegnet: Map learning with lane segment perception for autonomous driving, ICLR2024.
>
> [4] Stream query denoising for vectorized hd-map construction, ECCV2024.
>
> [5] Topologic: An interpretable pipeline for lane topology reasoning on driving scenes, NIPS 2024.

---

> ### Author Response · Authors · 2025-11-25
> **A Warm Reminder Regarding Our Previous Response**
>
> Dear Reviewer sabF,
>
>
> I hope you are doing well. I am writing to kindly follow up on my earlier response. If there is anything further you would like me to clarify or provide, please feel free to let me know. Thank you again for your time and effort in reviewing our submission. I look forward to your reply.
>
> Best regards,
>
>
> Authors of Submission 360

---

> ### Author Response · Authors · 2025-11-28
> **A Warm Reminder Regarding Our Previous Response**
>
> Dear Reviewer sabF,
>
> I hope you are doing well.
>
> I am writing to kindly follow up on our previous response. We hope our point-by-point rebuttal has adequately addressed the concerns you raised.
>
> Should any further clarification or additional information be needed, please do not hesitate to let me know.
>
> Thank you once again for your time and valuable effort in reviewing our submission . We look forward to hearing from you.
> Best regards,
>
> The Authors of Submission 360

---

### Official Review · Reviewer_Cnwo · 2025-11-01

**Soundness:** 3
**Presentation:** 3
**Contribution:** 3
**Rating:** 8
**Confidence:** 4

**Summary:**

This paper addresses the task of topological reasoning for autonomous driving and proposes a new model called TopoStreamer. The authors identify two key issues in current streaming-based learning: (1) consistent positional embedding and (2) temporal multiple attribute learning for lane segments. To tackle these challenges, they introduce dynamic explicit positional encoding, multiple streaming attribute constraints, and a lane segment denoising module.

Experimental results show that TopoStreamer significantly improves lane segment detection by better leveraging temporal information and also achieves notably higher accuracy in topology prediction. The authors also provide detailed visualizations to support their analysis. Overall, this is a solid and well-executed paper.

**Strengths:**

1. The authors rightly choose to incorporate temporal information into the lane segment topology reasoning task, as it helps address missed detections caused by occlusions and high-speed motion—a critical issue in topology reasoning.

2. The two challenges highlighted by the authors, consistent positional embedding and temporal multiple attribute learning for lane segments, are well-motivated and meaningful. Moreover, their proposal of a new metric to evaluate lane boundary classification accuracy is a valuable addition that further enhances the OpenLaneV2 benchmark.

3. The denoising module introduced in this work is also interesting and could serve as a useful foundation for future research.

**Weaknesses:**

1. I think the authors lack a detailed description of Section 3.4, LANE SEGMENT DENOISING. Why is denoising needed for predicting the topological relationships of these fine-grained lane segments? I hope the authors can further explain the motivation behind denoising, preferably with some simple examples.

2. Since introducing temporal information incurs significant computation, I hope the authors can provide the training and inference times.

**Questions:**

See Weakness

---

> ### Author Response · Authors · 2025-11-14
> **Response to Reviewer Cnwo**
>
> ### Author Response
>
> We sincerely thank the reviewers for their recognition of our work. Based on the valuable feedback provided, we have made substantial revisions to the manuscript. We kindly invite the reviewers to refer to our updated version when reviewing our responses. Below, we provide point-by-point answers to the raised questions:
>
> ---
>
> #### **1. Detailed description of lane segment denoising**
>
> The accuracy of topology prediction is highly dependent on the quality of lane detection. For instance, when noise causes misalignment between the endpoints of two lane segments that should be connected, it can significantly compromise the reliability of topological relationship inference.
> To address this issue, we introduce a **denoising mechanism** during training, enabling the model to learn from various noisy patterns. These noise patterns often lead to fragmented lane segments and positional shifts, thereby reducing the likelihood of correct topological associations. By learning to denoise, the model can recover the original positions and connectivity of lane segments, ultimately improving both detection robustness and topology inference performance.
>
> We have added relevant explanations in **Section 3.4**, with the new text **highlighted in red (lines 282–290)**.
> Furthermore, a concrete example of the denoising process is provided in **Figure 2** of the appendix, and the corresponding description in **Section A.2** is also **highlighted in red (lines 780–785)** for clarity.
>
> ---
>
> #### **2. Training and testing time**
>
> The training and inference times are related to the model and the quantity of GPUs used. Our setup utilizes **4 × V100 GPUs**, with a **training time of approximately 28 hours** and an **testing time of about 40 minutes**. Increasing the number of GPUs can accelerate the process—for instance, switching to **4 × A100 GPUs** reduces the training time to **around 20 hours**.
>
> Detailed information on training and inference times is provided in the **appendix, Section A.9**, with the new text **highlighted in red (lines 927–931)**.
> Additionally, to evaluate the extra computational cost compared with the baseline, we include an analysis of **computational complexity** in **Section A.10** of the appendix(**highlighted in blue, lines 933–943**).
> This additional cost leads to a significant and effective improvement in performance.

---

> > ### Comment · Reviewer_Cnwo · 2025-11-25
> >
> > The authors have addressed my concerns. They have also made substantial improvements to the paper during the rebuttal, and they now demonstrate the distinctiveness and effectiveness of their method much more clearly compared with existing approaches. Current lane topology reasoning methods do not consider the temporal consistency of lane segments or their topological relations. The proposed temporal road topology reasoning fills this gap. By focusing on the temporal perception of lane segments with multiple attributes, it provides valuable insights and brings positive implications for dense HD map construction and autonomous driving. I will maintain my high score of 8.

---

### Author Response · Authors · 2025-11-24
**Global Response to Area Chair**

We sincerely thank the Area Chair and the reviewers for their insightful comments and constructive feedback on our manuscript. Based on these valuable suggestions, we have made substantial revisions that significantly improve the clarity, technical depth, and experimental rigor of the paper. The key improvements are summarized below:

---

## **1. Enhanced Methodological Justification and Novelty**

- We clarified the motivation and novelty of the proposed **TopoStreamer** framework, emphasizing its temporal design tailored for learning **multiple attributes of lane segments**—including centerlines, boundaries, semantic types, and topology—which existing temporal mapping methods do not fully address.

- The manuscript now includes detailed comparisons and ablation experiments against **major prior works** such as StreamMapNet, MapQR, LaneSegNet, and SQD-MapNet (Tables **4a, 4b, 5**). These results highlight the strengths of our core contributions:
  - **Streaming Attribute Constraints**
  - **Dynamic Boundary Point Encoding**
  - Enhanced **Lane Segment Denoising** mechanism

---

## **2. Introduction of a Comprehensive Denoising Mechanism**

- We have expanded the description of our **lane segment denoising** process (Section **3.4**), which is essential for improving detection robustness and topological inference by learning to recover from noisy lane-segment patterns. An illustrative example has been added in the appendix (**Figure 2, Section A.2**).

- Our improved query denoising design jointly incorporates boundary and topology constraints, showing clear improvements over the baseline denoising method (**Table 4b**).

---

## **3. Extended Experimental Analysis**

- We performed extensive ablations validating every module in our framework. These results demonstrate:
  - **+0.9% mAP** and **+1.5% TOP_{lsls}** from our streaming constraints over a baseline transformation loss
  - **+1.8–2.1% mAP** from DBPE over alternative positional encoding approaches
  - **+1.1% mAP** and **+0.8% TOP_{lsls}** from our enhanced denoising module

- By integrating the topological reasoning module from **TopoLogic** into our system, we achieved an additional **+1.6% improvement in TOP_{lsls}**, reported in **Table 2**.

---

## **4. Improved Clarity and Presentation**

- We added a **nomenclature table (Table 1)** to unify and clarify all symbols used in the paper.

- Additional details on the **stream memory mechanism** are provided in **Appendix A.1**, and a typo in **Equation (4)** has been corrected.

- We clarified the hyperparameter selection strategy for our loss weights and explained its consistency with the LaneSegNet baseline (**Appendix, Lines 800–802**).

---

## **5. Analysis of Computational Complexity and Efficiency**

- The computational complexity analysis has been significantly expanded (**Appendix, Section A.10; Table 3**), comparing our method with LaneSegNet.

- Results show that **TopoStreamer** improves performance by **4.0% mAP** and **3.1% TOP_{lsls}** while introducing only a **marginal increase** in FLOPs (from 639.1G to 652.1G), demonstrating an excellent accuracy–efficiency trade-off.

---

## **6. Detailed Implementation and Training Specifications**

- We now explicitly report training time (28 hours on **4×V100 GPUs**) and inference time (40 minutes) in **Appendix A.9** to facilitate reproducibility.

---

## **7. Reviewers’ Responses to Our Rebuttal**

- **Reviewer Cnwo** maintained a highly positive evaluation, noting that our work fills an important gap in lane topology reasoning, where existing methods overlook temporal consistency of lane segments and their topological relations. The reviewer emphasized that our method provides valuable insights for dense HD-map construction and autonomous driving.

- **Reviewer sabF** initially had misunderstandings regarding our contributions. We addressed these concerns point-by-point, demonstrating that our modules are **specifically designed for multi-attribute lane-segment learning** and that **integrating them seamlessly into a unified end-to-end framework is both novel and effective**. Although the reviewer could not respond further due to time constraints, we believe our clarifications will lead to a positive reassessment.

- **Reviewer h3uG** also did not respond after the rebuttal deadline. However, our detailed responses fully resolved the concerns raised and significantly improved the overall manuscript quality. We are confident the reviewer will recognize these improvements in the final evaluation.

---

All revisions have been carefully integrated into the manuscript. Although some reviewers with initial misunderstandings were unable to respond before the rebuttal deadline, a reviewer explicitly expressed **strong positive evaluations** based on our rebuttal and acknowledged the improvements made.

We respectfully invite the Area Chair to consider these comprehensive enhancements when assessing our submission.

---

### Meta-Review · Area_Chair_FCZD · 2026-01-06

**Summary:**

The paper proposes TopoStreamer, a temporal framework for lane segment topology reasoning that introduces streaming attribute constraints, dynamic boundary point positional encoding (DBPE), and lane segment denoising. All reviewers agree that incorporating temporal modeling into lane segment perception is important and that the method achieves strong empirical performance, including state-of-the-art results on OpenLane-V2 for both lane segment and centerline perception. However, the reviewers differ in their assessment of novelty and contribution. Reviewer Cnwo believes that the revised paper demonstrates clear distinctiveness and fills a gap in temporal multi-attribute lane topology reasoning, whereas reviewers sabF and h3uG consider the method to largely build on existing temporal and HD-map learning frameworks, with improvements that are mainly incremental.

**Reviewer Concerns:**

In the rebuttal, the authors provided extensive ablations, explicit comparisons with StreamMapNet, MapQR, SQD-MapNet, and LaneSegNet, as well as computational complexity analysis, training and inference time, improved explanations of denoising, positional encoding, and stream memory, and corrections to notation and equations, which together addressed most concerns related to clarity, soundness, and experimental rigor. Nevertheless, the concern regarding novelty was not fully resolved. The AC agrees with reviewers sabF and h3uG that the proposed streaming constraints, boundary-based positional encoding, and denoising mechanisms are best interpreted as variations or extensions of existing temporal propagation, positional encoding, and query denoising techniques. While applying these components to lane segment topology reasoning is technically meaningful, the resulting contribution remains primarily an adaptation of existing ideas, and the level of conceptual innovation does not meet the expectations for a top-tier conference.

**Reviewer Scores:**

Reviewer Cnwo maintained a positive evaluation, with a score of 8 both before and after the rebuttal. Reviewers sabF and h3uG each gave a score of 2 initially and did not revise their assessments after the rebuttal, so their scores remained at 2. As a result, the paper received one accept and two rejects. Although the rebuttal and revisions substantially improved the clarity, experimental validation, and technical soundness of the work, the majority of reviewers remained unconvinced that the contributions go beyond well-executed extensions of existing temporal and HD-map learning methods to the level of conceptual novelty required for acceptance. Taking into account these unresolved concerns regarding novelty, the final recommendation is to reject the paper.

---

### Decision · Program_Chairs · 2026-01-26

Reject